# Inference and Learning in a Latent Variable Model for Beta Distributed Interval Data

**DOI:** 10.3390/e23050552

**Published:** 2021-04-29

**Authors:** Hamid Mousavi, Mareike Buhl, Enrico Guiraud, Jakob Drefs, Jörg Lücke

**Affiliations:** 1Machine Learning Lab, Department of Medical Physics and Acoustics and Cluster of Excellence Hearing4all, University of Oldenburg, 26129 Oldenburg, Germany; enrico.guiraud@cern.ch (E.G.); jakob.drefs@uni-oldenburg.de (J.D.); joerg.luecke@uni-oldenburg.de (J.L.); 2Medical Physics Group, Department of Medical Physics and Acoustics and Cluster of Excellence Hearing4all, University of Oldenburg, 26129 Oldenburg, Germany; mareike.buhl@uni-oldenburg.de; 3European Organization for Nuclear Research, (CERN), 1211 Meyrin, Switzerland

**Keywords:** latent variable models, Bayes nets, Beta distribution, noisy-OR, expectation maximization, variational inference

## Abstract

Latent Variable Models (LVMs) are well established tools to accomplish a range of different data processing tasks. Applications exploit the ability of LVMs to identify latent data structure in order to improve data (e.g., through denoising) or to estimate the relation between latent causes and measurements in medical data. In the latter case, LVMs in the form of noisy-OR Bayes nets represent the standard approach to relate binary latents (which represent diseases) to binary observables (which represent symptoms). Bayes nets with binary representation for symptoms may be perceived as a coarse approximation, however. In practice, real disease symptoms can range from absent over mild and intermediate to very severe. Therefore, using diseases/symptoms relations as motivation, we here ask how standard noisy-OR Bayes nets can be generalized to incorporate continuous observables, e.g., variables that model symptom severity in an interval from healthy to pathological. This transition from binary to interval data poses a number of challenges including a transition from a Bernoulli to a Beta distribution to model symptom statistics. While noisy-OR-like approaches are constrained to model how causes determine the observables’ mean values, the use of Beta distributions additionally provides (and also requires) that the causes determine the observables’ variances. To meet the challenges emerging when generalizing from Bernoulli to Beta distributed observables, we investigate a novel LVM that uses a maximum non-linearity to model how the latents determine means and variances of the observables. Given the model and the goal of likelihood maximization, we then leverage recent theoretical results to derive an Expectation Maximization (EM) algorithm for the suggested LVM. We further show how variational EM can be used to efficiently scale the approach to large networks. Experimental results finally illustrate the efficacy of the proposed model using both synthetic and real data sets. Importantly, we show that the model produces reliable results in estimating causes using proofs of concepts and first tests based on real medical data and on images.

## 1. Introduction

Automatic systems able to infer hidden causes from data are of considerable interest for the automation in a wide range of applications. In medical data analysis, for instance, an important task of medical doctors is to infer a possible set of causes from an array of patient symptoms. In this context, machine learning algorithms, specifically Latent Variable Models (LVMs) and noisy-OR Bayes nets, are well established tools. They aim at automatizing previous manual inference processes to further assist both patients and doctors. In recent years, machine learning algorithms have shown promising results in constructing a causal graph between the latents and observables and thus estimating a diagnostic model from a set of patient data [1,2,3,4]. Once trained, they can assign probabilities to hidden causes (e.g., diseases) given a set of observables (e.g., symptoms). Such models for medical data analysis, e.g., in the form of noisy-OR networks [2,5,6,7,8], assume medical symptoms to be encoded by binary observables (a symptom is present or absent). As an example of the studies in this direction, we refer the readers to [2] where three different probabilistic models (namely noisy-OR, naive Bayes nets and logistic regression) have been investigated in order to infer a knowledge graph from electronic health records. Amongst the three models, the authors have shown that a probabilistic noisy-OR model can outperform the other two and produce state-of-the-art results that are further used for diagnosis.

Nevertheless, it can be argued that assuming a binary representation is simplistic. Given a data set with records of measurements of symptoms, the distribution of symptoms can range from not present, over intermediate, to severe. Furthermore, for any given cause (any disease), the distribution of symptoms may be of interest for data analysis (e.g., for precise disease characterization). Indeed, many schemes for medical records taking establish a more continuous recording of symptoms [9,10,11]. If symptoms are binarized, as required for noisy-OR networks or as provided by data sets such as QMR-DT [12], then not all statistical information can be leveraged. Approaches that can infer causes from non-binarized symptoms may consequently be able to extract more information, which motivates our investigation of generalizations from Bernoulli to Beta distribution for observables.

Instead of further developing probabilistic data descriptions (as we do here), a potential alternative would be the use of deep neural network (DNN) approaches. So far, a number of DNNs including Variational Autoencoders (VAEs), Convolutional Neural Networks (CNNs) and Recurrent Neural Networks (RNNs) have been used for learning diagnostic models (see, e.g., [13,14,15,16,17,18] and references therein). In recent years, DNNs have shown a strong capability in analyzing medical and other advanced data sets and have consequently gained a considerable attention. This is importantly due to the availability of large data sets, e.g., in the form of electronic health records. To some extent, DNNs can outperform probabilistic models in terms of both accuracy and run-time as evaluation measures. However, it could be argued that they are not appropriate tools for medical reasoning in the context that we considered here due to the following reasons: First, the results obtained from DNNs are usually not interpretable as the taken assumptions and also the procedure of producing the results are relatively unclear [19,20]; we stress that interpretability is a substantial factor for medical data reasoning. Secondly, DNNs commonly provide a final result without refined representation, e.g., of alternative other likely solutions or of statistical properties of symptoms/causes [21]. Thirdly, DNNs usually require a large amount of clean data sets for training and, in some cases, have been observed to perform rather poorly in the absence of certain important information (see, e.g., [22] for a concrete comparison between different machine learning tools for medical data analysis including DNNs and probabilistic models). Taken together, more explicit probabilistic graphical models may, consequently, be preferable.

In this work, we aim at generalizations of standard noisy-OR models towards models with observable values that lie within an interval, as they occur, e.g., for medical data analysis. An example for medical data that we will use is provided in the form of Common Audiological Functional Parameters (CAFPAs) [9,23] describing certain audiological symptoms which can be used to infer different types of causes for hearing loss or hearing deficits. Another example that we will use is image data that are conventionally stored by assigning a pixel value in an interval (e.g., [0,255]). Motivated by such data and by the success of noisy-OR networks, we here seek a graphical model description of interval data using a bipartite graph as noisy-OR nets. We also maintain binary latents to model the presence or absence of a disease. However, we will use Beta distributions to model observables instead of Bernoulli distributions used by standard noisy-OR Bayes nets. The generalization from Bernoulli to Beta distributions instantly results in significant challenges that have to be addressed: First, when replacing binary {0,1} symptoms by an interval of values in [0,1], the corresponding distributions, e.g., the Beta distribution, has parameters for variable means as well as variable variances (while a Bernoulli distribution has one parameter that determines mean and variance). The more refined statistics of interval data therefore suggests for an LVM approach to model mean as well as variance combinations given a set of non-zero causes. As a second main challenge, we require a background model to avoid degeneracies for the absence of all causes (i.e., if all causes are zero). Thirdly, we require learning and inference to be scalable since the number of diseases and symptoms is (as for noisy-OR networks) potentially large, which renders exact inference and learning intractable.

Starting point for our investigation will be non-linear combinations of causes that are well suited for continuous observables. As noisy-OR combinations are specific to binary latents, we here use the maximum non-linearity as combination rule [24,25,26,27,28,29,30]. The maximum is (A) a relatively canonical choice, (B) a suitable model for the combination of, e.g., diseases, and (C) will turn out to be particularly convenient to model the combination of symptom variances. In previous work, the use of a maximum non-linearity for latent causes has been motivated by properties of acoustic data (e.g., [24]); and has repeatedly been used for LVMs with observables that are, e.g., distributed according to a Poisson distribution [28] as well as w.r.t. a Gaussian distribution [25,26,27,31,32].

Here we will exploit more recent analytical results that apply if the maximum combination is used to combine binary causes. More concretely, we will use a theorem derived by [33], which shows how learning equations can in principle be derived if observables follow a distribution of the exponential family. By far most previous LVM approaches have focused on Gaussian observables [25,26,31,34,35,36] including LVMs that have used the maximum combination instead of the more conventional linear sum [25,26,31]). However, there have also been a number of LVM approaches that aimed as [33] at deriving results for distributions other than Gaussians. For instance, exponential family sparse coding (EF-SC) [37], exponential family probabilistic component analysis (EF-PCA) [38] or Bayesian exponential family PCA [39,40] all considered distributions of the exponential family. Except of [33], in all the other studies, generalizations of standard linear summations for Gaussian observables were the starting point towards the generalization to exponential families. Importantly for this work, none of the previous LVM approaches has shown realizations of their approaches for more intricate distributions of the exponential family (such as the Beta distribution). Any experimental results (and usually the corresponding analytical derivations) of previous work [37,38,39,40] were restricted to one-parameter distributions of the exponential family (such as Bernoulli and Poisson). To our best knowledge, details of how their theoretical results can be generalized to incorporate more intricate probability distributions (such as Beta) have not been discussed (not in their original paper nor in related studies). The exception is represented by [33] who provide results for the two-parameter Gamma distribution. Neither [33] nor any of the other above discussed LVM models has shown analytical or experimental results for Beta distributed observables, however.

To continue our discussion of related work, studies like [41,42] suggested an automatic procedure using a fully Bayesian method to estimate the statistical dependencies from a set of heterogeneous data. In particular, the method introduced by Valera and Ghahramani in [41] can successfully model the true statistical types of data as being real-valued, positive real-valued, interval, categorical or ordinal, and defines a mixture of likelihood functions that factorizes for each of the considered data types. However, as a Markov chain Monte Carlo (MCMC) algorithm is used for training, application of the model at large-scales is relatively costly. The work by Vergari et al. [42] further generalizes this approach so that the new model can also robustly estimate missing values, corruption and anomalies in the data. The new approach, known as Automatic Bayesian Density Analysis (ABDA), then shows a consistent performance for automatic exploratory analysis of complex data.

In Section 2, we will motivate the maximum combination (together with other prerequisites of the model) using the example of diseases and symptoms modelling for medical data. Because of its actuality we specifically consider COVID-19 as an introductory example but will use medical data of hearing impairments [9] for numerical experiments in Section 5 as such data is directly at our perusal. The symptoms will be assumed to be continuous and to lie in a finite interval. We will later present our generative model in Section 3 and discuss the specifics of using a Beta distribution to model such interval data. In Section 4, we will apply expectation maximization (EM) [43] to train our model, and we will use the theoretical results presented in [33] to obtain a set of update equations. We will specifically apply variational EM approximations in order to scale the novel model to realistically large model sizes. Similar to previous approximations for models with binary latents [28,44], we use truncated posteriors. Unlike earlier approaches however, we apply a novel and fully variational form of such approximations [45,46,47]. Further, Section 5 will present the first results that we obtained using both synthetic and real data sets, and finally in Section 6, we will conclude the paper.

## 2. A Preliminary Maximal Causes Model with Beta Observables

Let us first consider a relatively straightforward but preliminary generative model that generalizes binary observables to observables with values in unit intervals. For this, consider the maximal causes generative model [28] (MCA) which uses the maximum in place of a linear summation as model for the combination of causes. Let $\vec{y}=\left(y_{1},\ldots ,y_{D}\right)$ be a *D*-dimensional vector in interval ${\left[0,1\right]}^{D}$. Then, by simply using a Beta distribution instead of a Gaussian, a preliminary version of the corresponding generative model for interval data can be defined as follows: (1)$$\begin{array}{cc} p(\vec{s}|\Theta ) & =\prod_{h=1}^{H}\mathrm{Bernoulli}\left(s_{h};\pi_{h}\right),\;s_{h}\in \left\{0,1\right\} \end{array}$$(2)$$\begin{array}{cc} p(\vec{y}|\vec{s},\Theta ) & =\prod_{d=1}^{D}\mathrm{Beta}\left(y_{d};{\bar{\mu }}_{d}\left(\vec{s},\Theta \right),\sigma^{2}\right),\;y_{d}\in \left[0,1\right] \end{array}$$(3)$$\begin{array}{cc} \mathrm{where}\;\;\;\;{\bar{\mu }}_{d}\left(\vec{s},\Theta \right) & =\max_{h}\left\{s_{h}M_{dh}\right\},\;h=1,\ldots ,H\;\mathrm{and}\;d=1,\ldots ,D \end{array}$$
where the Beta distribution is parameterized w.r.t. the distribution mean and a distribution variance $\sigma^{2}$. For the given *H* hidden variables, the Bernoulli distribution assumes a cause to be present or not ($s_{h}\in \left\{0,1\right\}$) and assigns a prior probability $\pi_{h}\in \left[0,1\right]$ to each cause *h*. Moreover, the model has as parameters the prior probabilities $\pi_{1},\ldots ,\pi_{H}$, the variance $\sigma^{2}$ (which is assumed to be the same for all Beta distributions), and a D×H weight matrix *M* with elements in interval (0,1) that models the dependency between latents and observables. Each vector ${\vec{M}}_{h}=\left(M_{1h},\ldots ,M_{Dh}\right)$ is referred to as a basis function or generative field. The set of all model parameters is consequently $\Theta =(\vec{\pi },M,\sigma^{2})$. The hidden variables (causes) $s_{h}$ determine the *mean* of an observable *d* through a maximum function: For a given observable *d* the active cause with the largest value of $M_{dh}$ determines the mean. An alternative combination model could, for instance, use a linear summation followed by a sigmoidal function [48,49] (SBN). The noisy-OR combination can not be used, however, as it is defined for binary observables.

The data model (Equation 1)–(Equation 3) generalizes the previous MCA models [26,27,28,31,32] by replacing the Gaussian or the Poisson distribution by the Beta distribution. Consequently, the model can cope with interval data and would be considered as a generalization of non-linear binary data models (e.g., noisy-OR). At a closer inspection, however, the considered model may still require some further generalizations that are preeminent for analyzing medical data. One example generalization which we will require further below concerns the global variance modelling. Already in the Gaussian case, it could be argued in favor of a variance that also depends on the causes (see, e.g., [50]). For the Beta distribution, a global fixed variance can cause more severe problems. In the following, we develop some intuition using the example of diseases and symptoms modelling in order to discuss the required generalizations of our data model (Equation 1)–(Equation 3).

As an example, here, consider the outbreak of COVID-19 and the difficulty to distinguish the novel disease from other diseases such as “common cold” or “flu” which cause similar symptoms. For simplicity, let us assume four medical symptoms: “fever”, “cough”, “sneezing” and “fatigue”. For the given set of diseases and symptoms, a binary representation of the causes/symptoms relations assigns 0 to the cases where the symptom is not present (i.e., the corresponding measurement is similar to the one of a healthy patient) and 1 otherwise.

In contrast to modelling a symptom as being present or absent, the LVM (Equation 1)–(Equation 3) assumes continuous symptoms, i.e., a symptom is modelled to range from not or barely present (close to zero) over intermediate (around 0.5) to very severe (close to one). This graded modelling is different from conventional binary or ordinal encodings and can be considered as a generalization of both cases. Figure 1 depicts an example of binary vs. continuous representation of the given set of diseases/symptoms (we refer to it as the disease profile). Note that the presented disease profile is, however, only an example of how such diseases and symptoms could be related to each other, and the figure may not be accurate (we refer the readers, e.g., to [51,52] for more details regarding the given causes/symptoms relations). Nevertheless, the disease profile illustrates how a continuous encoding, in contrast to a binary encoding, enables us to capture higher order statistics of the observables and to model more information of the corresponding symptoms.

In particular, each disease causes different symptoms, e.g., “COVID-19” reliably causes “cough” and a strong “fever”. In the Bayes net (Equation 1)–(Equation 3), the relation of diseases to the symptoms is modelled by the weight matrix *M* where each element $M_{dh}$ distinguishes the strength connection between cause (disease) *h* and observation (symptom) *d*. For instance, the weight $M_{dh}$ modelling the connection from “COVID-19” to “fever” is strong while the weight $M_{dh}$ modelling the connection from “COVID-19” to “fatigue” is rather weak, and so forth (here strong means a value closer to 1 while weak means closer to 0). Now, suppose that only the “COVID-19” disease is active, then it sets the corresponding symptom for “fever” to a value close to one. Consequently, the maximum combination (Equation 3) sets the mean of observables to be close to one. The Beta distribution then models that the actual value can be a bit higher than the mean or a bit lower (but never outside the range [0,1]).

Moreover, causes (diseases) have different probabilities to occur (different $\pi_{h}$ values). That is, in some regions, for instance, contracting to “COVID-19” is very likely such that the prior $\pi_{h}$ for “COVID-19” is rather high while the prior for “common cold” or “flu” could be low. Importantly, causes (diseases) can co-occur, which highlights the importance of a possible combination of the causes which we modelled here using the maximum (Equation 3). For any one symptom, a plausible model for symptom combination is to assume the symptom mean to be set by the disease with the strongest influence on the symptom. If the strongest disease is, for instance, always causing a “fever” then this small variance should also apply for the symptom “fever” even though other diseases are active. A disease with an intermediate probability for “fever” and a high variance (like “common cold”) should not make the variance for “fever” large if, e.g., “COVID-19” is active. In this case, however, the model (Equation 1)–(Equation 3) generates the symptoms for “fever” using the mean value equals to the $M_{dh}$ connecting “COVID-19” to “fever” (the strongest connection) and a global variance $\sigma^{2}$ that results in a mild “fever”. This ascertains one main limitation of the preliminary model (Equation 1)–(Equation 3) where a global variance encoding of the observables is assumed. Therefore, we will later motivate a more general model that assumes a generalization of the global variance parameter of the model and thus alleviates the effect of this issue.

Another limitation of the LVM model (Equation 1)–(Equation 3) is the case where none of the causes (diseases) are active. Consequently, the mean of observables in (Equation 3) will be obtained as zero which is not valid for the Beta distribution (the mean of Beta distribution must be in the interval (0,1)). To address this issue, in addition to the *H* considered hidden causes for diseases, we further assume a background model for our generalization which combines with the corresponding active causes and provides (if all $s_{h}=0$) non-zero mean and variance values. The background model thus prevents degeneracies and models, in the case of medical data, the symptom statistics of healthy patients. We denote this cause by $s_{0}=1$ that is much like how bias terms are usually modelled. Given latent variable $\vec{s}$, we then demand the mean of observable $y_{d}$ to compute by:(4)$$\begin{array}{c} {\bar{\mu }}_{d}\left(\vec{s},\Theta \right)=\max_{h}\left\{s_{h}M_{dh}\right\},\;h=0,\ldots ,H\;\mathrm{and}\;d=1,\ldots ,D \end{array}$$
where the basis function ${\vec{M}}_{0}$ contains background parameters and corresponds to $s_{0}=1$.

Applying the two generalizations mentioned above is a formidable challenge that, to our best knowledge, has not been considered previously. While the dominance of one cause is modelled by the maximum (Equation 3) as for previous MCA approaches, modelling the dominance of the same cause also for the variance has not been used neither for maximum combinations nor for linear or any other combination models. The exception is a sparse coding model with Gaussian observables [50] as well as the general (and above discussed) theoretical treatment for exponential family distributions [33]. In the following, the theoretical results of the latter will be used to obtain an LVM model for Beta distributed observables which overcomes the limitations of the preliminary model (Equation 1)–(Equation 3).

## 3. A Complete LVM Model with Beta Distributed Observables

In order to utilize theoretical results of [33] for the aimed generalizations, note that exponential family distributions with two parameters can in general be parameterized differently, e.g., by using standard shape parameters (like for Beta or Gamma), mean and variance parameters (μ, $\sigma^{2}$), natural parameters ($\eta_{1}$, $\eta_{2}$), or by using mean value parameters ($w_{1}$, $w_{2}$) [53]. In the case of the Beta distribution, the standard shape parameters α and β are the same as the natural parameters $\eta_{1}$ and $\eta_{2}$. Using the natural parameters, the mean value parameters of the Beta distribution are defined as follows:(5)$$\begin{array}{c} w_{1}:=\;{\langle \log\left(y\right)\rangle }_{p}\;\mathrm{and}\;w_{2}:=\;{\langle \log\left(1-y\right)\rangle }_{p}, \end{array}$$
where ${\langle f\left(y\right)\rangle }_{p}$ represents the expected value of a function f(y) w.r.t. a Beta distribution $p(y\;|\;\eta_{1}$, $\eta_{2})$ with natural parameters $\eta_{1}$ and $\eta_{2}$. The following relations then exist between the mean value parameters, the natural parameters, and the mean and variance parameters of the Beta distribution:(6)$$\begin{array}{c} \mu =\frac{\eta_{1}}{\eta_{1}+\eta_{2}}\;\mathrm{and}\;\sigma^{2}=\frac{\eta_{1}\eta_{2}}{{\left(\eta_{1}+\eta_{2}\right)}^{2}\left(\eta_{1}+\eta_{2}+1\right)} \end{array}$$
and
(7)$$\begin{array}{c} w_{1}=\psi \left(\eta_{1}\right)-\psi \left(\eta_{1}+\eta_{2}\right)\;\;\mathrm{and}\;\;w_{2}=\psi \left(\eta_{2}\right)-\psi \left(\eta_{1}+\eta_{2}\right), \end{array}$$
where ψ(.) is the Digamma function [54]. Moreover, in the case that $\eta_{1}\neq \eta_{2}$, there exists a bijective mapping $\vec{\Phi }$ such that:(8)$$\begin{array}{c} \vec{\Phi }\left(\vec{w}\right)=\vec{\eta }\;\mathrm{with}\;\vec{w}=\left(w_{1},w_{2}\right),\;\vec{\eta }=\left(\eta_{1},\eta_{2}\right). \end{array}$$

Observe that the assumption of $\eta_{1}\neq \eta_{2}$ is required for the existence of function $\vec{\Phi }$ (compare, e.g., [53,55] under the name of minimal representation property of the exponential family distributions).

Now, consider *N* conditionally independent and Beta distributed observed variables $Y=\{{\vec{y}}^{\left(1\right)},\ldots ,{\vec{y}}^{\left(N\right)}\}$ where each input ${\vec{y}}^{\left(n\right)}=\left(y_{1}^{\left(n\right)},\ldots ,y_{D}^{\left(n\right)}\right)$ is a *D*-dimensional vector in interval ${\left[0,1\right]}^{D}$. Further let the Beta distribution to be parameterized w.r.t. its mean value parameters. Then we generalize the preliminary generative model (Equation 1)–(Equation 3) as follows: (9)$$\begin{array}{cc} p(\vec{s}|\Theta ) & =\prod_{h=0}^{H}\mathrm{Bernoulli}\left(s_{h};\pi_{h}\right),\;\;s_{h}\in \left\{0,1\right\}\;\mathrm{and}\;\pi_{h}\in \left[0,1\right] \end{array}$$(10)$$\begin{array}{cc} p(\vec{y}|\vec{s},\Theta ) & =\prod_{d=1}^{D}\mathrm{Beta}\left(y_{d};{\bar{W}}_{d}\left(\vec{s},\Theta \right),{\bar{V}}_{d}\left(\vec{s},\Theta \right)\right),\;\;y_{d}\in \left[0,1\right] \end{array}$$(11)$$\begin{array}{cc} \mathrm{where}\;\;\;\;{\bar{W}}_{d}\left(\vec{s},\Theta \right) & =W_{dh(d,\vec{s},\Theta )},\;d=1,\ldots ,D \end{array}$$(12)$$\begin{array}{cc} {\bar{V}}_{d}\left(\vec{s},\Theta \right) & =V_{dh(d,\vec{s},\Theta )},\;d=1,\ldots ,D \end{array}$$(13)$$\begin{array}{cc} h(d,\vec{s},\Theta ) & ={\mathrm{argmax}}_{h}\left\{s_{h}M_{dh}\left(\Theta \right)\right\},\;h=0,\ldots ,H\;\mathrm{and}\;d=1,\ldots ,D \end{array}$$
where $\Theta =(\vec{\pi },W,V)$ represents the set of all parameters: *W* and *V* are matrices with D×(H+1) entries and $\vec{\pi }=\left(\pi_{0},\ldots ,\pi_{H}\right)$, where the index 0 corresponds to the background (we set $s_{0}=1$). Likewise, the weight matrix M(Θ) with D×(H+1) entries corresponds to the mean of observables, but this time M(Θ) is dependent to the parameters Θ of the model (we will later elaborate how this matrix will be obtained). For a given set of active causes, i.e., those taking the value $s_{h}=1$ including the background, we take the distribution of $y_{d}$ to be determined by the strongest connection (given the weight matrix M(Θ)) associated with the active causes and $y_{d}$. The index for this strongest connection is computed using the *argmax* function in (Equation 13) (note the relation but also the difference to the *max* function in (Equation 3)). Equation (Equation 13) together with Equations (Equation 11) and (Equation 12) then define the desired maximum superposition. Further, the maps ${\bar{W}}_{d}\left(\vec{s},\Theta \right)$ and ${\bar{V}}_{d}\left(\vec{s},\Theta \right)$ set the mean values of the Beta distribution of observable $y_{d}$ according to (Equation 11) and (Equation 12).

The definitions of the LVM model (Equation 9)–(Equation 13) are rather technical here but will enable us to optimally generalize our preliminary model (Equation 1)–(Equation 3). Such a generalization results in learning two dictionaries: *W* and *V*. Having these two dictionaries and further assuming the existence of the function $\vec{\Phi }$ in (Equation 8) (we have to assume $\eta_{1}\neq \eta_{2}$), we can compute the natural parameters as well as the mean and variance parameters of the Beta distribution. The use of mapping $\vec{\Phi }$ from mean value parameters to natural parameters is not only required for the formal definition of the generative model, but also the mapping will be required for the concrete parameter updates (by using Equations (Equation 18)–(Equation 20) further below). These updates will be formulated in terms of the mean value parameters. At the same time, the updates will also require expectation values w.r.t. the posteriors which are computable in closed-form for the natural parameters. Additionally, note (see below) that the mean for Equation (Equation 13) is given by a closed-form expression of the natural parameters. Now, in order to concretely compute the mean (and the variance) parameter of the Beta distribution, let Σ(Θ) be a D×(H+1) matrix containing, for each cause *h* (plus background), a set of *D* different standard deviations (one for each observable). Then, for d=1,…,D and h=0,…,H, we define:(14)$$\begin{array}{cc} M_{dh}\left(\Theta \right) & :=\frac{\Phi_{1}\left(W_{dh},V_{dh}\right)}{\Phi_{1}\left(W_{dh},V_{dh}\right)+\Phi_{2}\left(W_{dh},V_{dh}\right)} \end{array}$$(15)$$\begin{array}{cc} \Sigma_{dh}^{2}\left(\Theta \right) & :=\frac{\Phi_{1}\left(W_{dh},V_{dh}\right)\;\Phi_{2}\left(W_{dh},V_{dh}\right)}{{\left(\Phi_{1}\left(W_{dh},V_{dh}\right)+\Phi_{2}\left(W_{dh},V_{dh}\right)\right)}^{2}\left(\Phi_{1}\left(W_{dh},V_{dh}\right)+\Phi_{2}\left(W_{dh},V_{dh}\right)+1\right)}. \end{array}$$

Equations (Equation 14) and (Equation 15) taken together with Equations (Equation 11)–(Equation 13) model how mean and variance parameters of each cause combine to generate the statistics of the symptoms. The dominant cause per observable is obtained using the *argmax* function in (Equation 13). The function $h(d,\vec{s},\Theta )$ determines the effective weight on observable $y_{d}$ and further determines the mean value parameters and consequently the variance of the Beta distribution. As for the previous MCA models, each observable $y_{d}$ can have a different cause as the dominant cause. Here the dominant cause not only determines the observables’ mean but also its variance.

The proposed generative model (Equation 9)–(Equation 13) can be considered as a generalization of binary–binary models such as noisy-OR or Sigmoid Belief Networks [48,49] (SBN) towards continuously distributed interval data. The model takes advantage of the previous MCA approaches but decisively amends the original model in aspects central to our goals that are: (a) modelling interval data by using Beta distributions, and (b) exploiting properties of the maximum that allows to model a combination of symptom variances alongside with a combination of symptom means. Our example of medical data analysis may have motivated the necessity of modelling variance combinations sufficiently obvious. We addressed the dependence of the variance on the causes using one dominating cause per symptom, and the *argmax* function (Equation 13) determines the dominant cause. Additionally, note that given a data point ${\vec{y}}^{\left(n\right)}$, the dominant cause can be different for each *d*. We will refer to the proposed data model (Equation 9)–(Equation 13) as *Beta-MCA* and, in the following, seek the optimal parameters $\Theta^{*}$ that maximize the log-likelihood of the data given by:(16)$$\begin{array}{c} L\left(\Theta \right)=\log\left(p({\vec{y}}^{\left(1\right)},\ldots ,{\vec{y}}^{\left(N\right)}|\Theta )\right)=\sum_{n}\log\left(p({\vec{y}}^{\left(n\right)}|\Theta )\right)=\sum_{n}\log\left(\sum_{\vec{s}}p\left({\vec{y}}^{\left(n\right)},\vec{s}|\Theta \right)\right). \end{array}$$

## 4. Maximum Likelihood

We apply the Expectation Maximization (EM) algorithm to optimize the parameters Θ under the generative model described in (Equation 9)–(Equation 13). Following, e.g., [43], instead of maximizing the log-likelihood function, the so-called free energy function (aka. ELBO) F(q,Θ) that is a lower bound of the log-likelihood is maximized in two alternating E- and M-steps:(17)$$\begin{array}{cc} F(q,\Theta )= & \sum_{n}\sum_{\vec{s}}q^{\left(n\right)}\left(\vec{s}\right)\left\{\sum_{d}\log\left(p(y_{d}^{\left(n\right)};{\bar{W}}_{d}\left(\vec{s},\Theta \right),{\bar{V}}_{d}\left(\vec{s},\Theta \right))\right)+\sum_{h}\log\left(p(s_{h}|\Theta )\right)\right\}+ℋ\left(q\right) \end{array}$$
where ℋ(q) is the Shannon entropy and $q^{\left(n\right)}\left(\vec{s}\right)$ is a variational distribution w.r.t. the state parameters $\vec{s}$. The objective F(q,Θ) will be increased w.r.t. the $q^{\left(n\right)}\left(\vec{s}\right)$ in the E-step (while Θ is kept fixed) and w.r.t. the parameters Θ in the M-step (with fixed $q^{\left(n\right)}\left(\vec{s}\right)$). These steps monotonically increase the free energy and, in practice, also the log-likelihood of the data till convergence. Practically, the E-step maximizes the free energy by setting $q^{\left(n\right)}\left(\vec{s}\right)$ equal to the posteriors (we refer to this as the full EM; $q^{\left(n\right)}\left(\vec{s}\right)=p\left(\vec{s}\;|{\vec{y}}^{\left(n\right)},\Theta \right)$). In addition, the M-step maximizes the free energy w.r.t. the parameters Θ. In this manner, the derivatives of F(q,Θ) w.r.t. each of the parameters W,V and $\vec{\pi }$ is set to zero and the corresponding update equations are obtained accordingly.

### 4.1. Parameter Update Equations (M-Step)

For standard linear LVMs with Gaussian observables, parameter update equations can usually be derived in closed-form [56]. In case of non-Gaussian or non-linear LVMs the derivation of suitable parameter updates is a central challenge [26,28,33]. Such challenges are already significant if one-parameter distributions are assumed for observables. Therefore, the Beta distribution seems to be a choice that is particularly difficult to address. Because of the maximum used to combine causes, we can however leverage a result of Mousavi et al. [33] which applies for the whole exponential family. More concretely, we can apply an analytic expression for a necessary condition at free energy maxima. The expression (Theorem 1 of Mousavi et al. [33]) contains the sufficient statistics of the corresponding distribution, the data and an index matrix. Such analytical expressions can be used here for parameter updates of the Beta distribution (with its sufficient statistics log(y) and log(1−y)) and are given by: (18)$$\begin{array}{cc} W_{dh}^{\mathrm{new}} & =\frac{\sum_{n=1}^{N}{\langle A_{dh}\left(\vec{s},\Theta^{\mathrm{old}}\right)\rangle }_{q^{\left(n\right)}}\log\left(y_{d}^{\left(n\right)}\right)}{\sum_{n=1}^{N}{\langle A_{dh}\left(\vec{s},\Theta^{\mathrm{old}}\right)\rangle }_{q^{\left(n\right)}}} \end{array}$$(19)$$\begin{array}{cc} V_{dh}^{\mathrm{new}} & =\frac{\sum_{n=1}^{N}{\langle A_{dh}\left(\vec{s},\Theta^{\mathrm{old}}\right)\rangle }_{q^{\left(n\right)}}\log\left(1-y_{d}^{\left(n\right)}\right)}{\sum_{n=1}^{N}{\langle A_{dh}\left(\vec{s},\Theta^{\mathrm{old}}\right)\rangle }_{q^{\left(n\right)}}} \end{array}$$(20)$$\begin{array}{cc} \pi_{h}^{\mathrm{new}} & =\frac{1}{N}\sum_{n=1}^{N}{\langle s_{h}\rangle }_{q^{\left(n\right)}} \end{array}$$
where d=1,…,D and h=0,…,H, and where $q^{\left(n\right)}=q^{\left(n\right)}\left(\vec{s};\Theta^{\mathrm{old}}\right)$ depends on the old parameters. The index set $A_{dh}\left(\vec{s},\Theta \right)$ is given by:(21)$$\begin{array}{c} A_{dh}\left(\vec{s},\Theta \right):=\left\{\begin{array}{cc} 1\; & h=h(d,\vec{s},\Theta )\\ 0\; & otherwise \end{array}\right. \end{array}$$
with $h(d,\vec{s},\Theta )$ defined in (Equation 13).

Observe that Equations (Equation 18) and (Equation 19) are fixed-point updates that at their equilibrium points provide the optimal solutions of *W* and *V*. For the EM algorithm, we initialize parameters Θ of Beta-MCA and update each of the parameters using (Equation 18)–(Equation 20) in the M-step (we will perform each of the above updates only once for our experiments). The alternating E- and M-steps will continue until parameters Θ have sufficiently converged. The (locally) optimal value of $\Theta^{*}$ is then obtained at the convergence point. Note that we still require the function $\vec{\Phi }\left(W,V\right)$ in order to compute matrices M(Θ) and Σ(Θ). To this, one can use equations presented in (Equation 7) and approximate the Digamma function to obtain a relation between matrices *W* and *V* and the natural parameters. We therefore apply a straightforward method and solve the following system of non-linear equations w.r.t. D×(H+1) matrices $\eta_{1}$ and $\eta_{2}$ in each M-step:(22)$$\begin{array}{c} \left\{\begin{array}{c} W-\psi \left(\eta_{1}\right)+\psi \left(\eta_{1}+\eta_{2}\right)=0\\ V-\psi \left(\eta_{2}\right)+\psi \left(\eta_{1}+\eta_{2}\right)=0 \end{array}\right. \end{array}$$

We will then let $\Phi_{1}\left(W,V\right)=\eta_{1}$ and $\Phi_{2}\left(W,V\right)=\eta_{2}$ in (Equation 14) and (Equation 15) to compute matrices M(Θ) and Σ(Θ) corresponding to the mean and variance parameters of the Beta distributions. The obtained weight matrix M(Θ) can then be used to distinguish the dominant cause per observable in (Equation 13).

The updates (Equation 18)–(Equation 20) together with (Equation 21) and (Equation 22) provide (together with a full E-step) an EM method that can be used to train interval data using the proposed Beta-MCA model. Algorithm 1 finally presents a brief summary of the full EM algorithm presented here. In the next section, we will further assess the reliability of the proposed generative model together with its update equations using both artificial and real data sets. Before that, however, we should stress that computing the posteriors at larger scales will become infeasible, which limits the application of the model to small size data sets. Specifically, computations will increase exponentially with number of latents, *H*, as the summation in the posterior goes over all possible states $\vec{s}$ (we denote such a set by S which includes all different permutations of a bit vector, i.e., $\left|S\right|=2^{H}$). For larger *H*, we therefore exploit variational approximations (variational E-steps) in the form of truncated variational EM.
**Algorithm 1** Full EM for parameter updates of the Beta-MCA model
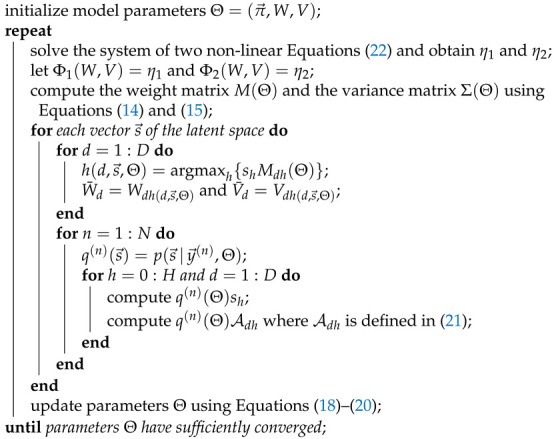


### 4.2. Truncated Approximations (Variational E-step)

We will apply the approach presented by, e.g., [45,46,57], that approximates full posteriors by truncating sums over the whole latent space to sums over those subsets which accumulate most of the posterior mass. In detail, we use truncated posteriors as approximation which are of the form:(23)$$q^{\left(n\right)}\left(\vec{s}|{\vec{y}}^{\left(n\right)},\Theta ,K\right):=\frac{p(\vec{s},{\vec{y}}^{\left(n\right)}\;|\;\Theta )}{\sum_{{\vec{s}}^{\prime }\in K^{\left(n\right)}}p\left({\vec{s}}^{\prime },{\vec{y}}^{\left(n\right)}\;|\;\Theta \right)}\delta \left(\vec{s}\in K^{\left(n\right)}\right)$$
where $\delta (\vec{s}\in K^{\left(n\right)})$ is 1 if $\vec{s}$ is in the subset $K^{\left(n\right)}$ and zero otherwise. The set of all $K^{\left(n\right)}$ is denoted by K, i.e., $K=(K^{\left(1\right)},\ldots ,K^{\left(N\right)})$. There is one subset $K^{\left(n\right)}$ for each data point ${\vec{y}}^{\left(n\right)}$, and it contains those states estimated to best describe the generation of ${\vec{y}}^{\left(n\right)}$ given the parameters Θ (see, e.g., [45,57] for details). Using the variational distributions (Equation 23), we further compute the following expected values for a well-behaved function $g(\vec{s})$:(24)$$\begin{array}{c} {\langle g\left(\vec{s}\right)\rangle }_{q^{\left(n\right)}}=\frac{\sum_{\vec{s}\in K^{\left(n\right)}}p\left(\vec{s},{\vec{y}}^{\left(n\right)}\;|\;\Theta \right)g\left(\vec{s}\right)}{\sum_{{\vec{s}}^{\prime }\in K^{\left(n\right)}}p\left({\vec{s}}^{\prime },{\vec{y}}^{\left(n\right)}\;|\;\Theta \right)}. \end{array}$$

The subsets $K^{\left(n\right)}$ can be chosen to be relatively small, which allows for training a Beta-MCA model with relatively high values of *H*. The main challenge, however, is to compute the subset $K^{\left(n\right)}$ (for each *n*) in E-steps such that the presented distributions in (Equation 23) represent a good approximation of the full posteriors. According to [45], these sets can be updated, e.g., using evolutionary algorithms with fitness defined as a monotonic function of the joint $p(\vec{s},\vec{y}\;|\;\Theta )$. This approach, termed Evolutionary Expectation Maximization (EEM), is black-box, i.e., it does not require any further derivations. Knowing the joint probability of a given generative model with binary latents is sufficient for the method to be applied. We can consequently directly use the EEM approximation to optimize the Beta-MCA model and obtain an efficient variational EM algorithm applicable also at larger scales.

Ideas of truncation are commonly used to optimize graphical models. In our context, ‘truncation’ originally referred to a truncation of the sums over states [31], or equivalently to a truncation of the full posterior to subset of the latent space [57]. Other types of truncated approximations used for efficient optimization of graphical models are mixtures of truncated exponentials (MTEs, e.g., [58,59]), where ‘truncation’ refers to exponential functions truncated to subspaces of their respective domain. For MTEs truncation is used to approximate the variables of the graphical model themselves (and complex graphical models can be optimized efficiently). The here used approach is complementary, as it uses truncation to approximate the posteriors resulting from the specific graphical model of Section 3.

## 5. Experiments

In the following, we will first consider the artificial bars test [60] to validate the efficacy of the presented update equations in finding ground-truth parameters. Next, medical data of hearing impairments is analyzed and the performance of the proposed Beta-MCA model in estimating the causes/symptoms relations is assessed in comparison to the noisy-OR model. Finally, we demonstrate the scalability of the Beta-MCA model by considering two further application domains of the model: Feature extraction and denoising. For these two experiments, we specifically leverage truncated posteriors (Equation 23) as variational distributions [45] (EEM).

### 5.1. The Bars Test

The bars test [28,60,61,62,63] is a well-known task for non-linear component extraction that is used to validate unsupervised learning algorithms. To generate data for this task, we assume H=10 basis functions ${\vec{M}}_{h}$ in the form of horizontal and vertical bars. Here we use continues values of 0.9 for each observable representing a bar (and a bar occupies 5 observables/pixels in a D=5×5 image) and 0 for other observables (non-bar pixels). Together with these basis functions, a background ${\vec{M}}_{0}$ with pixel values of 0.08 and 0.22 (a checkerboard) is assumed that ensures the lowest value of the observation mean to be greater than zero (see Figure 2B). Moreover, we here deliberately choose two basis functions with the same mean values but with different variances (causes 1 and 5 in Figure 2B) in order to assess the ability of the model in distinguishing between the two causes. Such a task is formidable for most of the previously established generative models as they learn a global variance parameter. In contrast, the proposed Beta-MCA model (Equation 9)–(Equation 13) is capable of training component variances alongside with the component means.

For the variance values ${\vec{\Sigma }}_{h}$, we consider different bar-like patterns with pixel values ranging from 0.1 to 0.2. The ground-truth patterns for variances were deliberately chosen to be different from the mean patterns (to better illustrate the modelling capabilities of the model). The background ${\vec{\Sigma }}_{0}$ was chosen to also (like the mean) have a checkerboard shape (with low pixel values of 0.01 and 0.05) to distinguish the background variance pattern from other patterns. Regarding the prior, we consider each of the causes (except the background $\pi_{0}$) to be active independently with a probability of $\pi_{h}=0.2$ (meaning two bars, on average, are active for each data point). The causes are then non-linearly superimposed according to the maximum function defined in (Equation 11)–(Equation 13). Note that for d=1,…,D, the mean $\bar{\mu }\left(\vec{s},\Theta \right)$ and variance ${\bar{\sigma }}^{2}\left(\vec{s},\Theta \right)$ of the Beta distribution are defined as follows:(25)$$\begin{array}{c} {\bar{\mu }}_{d}\left(\vec{s},\Theta \right)=M_{dh(d,\vec{s},\Theta )}\left(\Theta \right)\;\mathrm{and}\;{\bar{\sigma }}_{d}^{2}\left(\vec{s},\Theta \right)=\Sigma_{dh(d,\vec{s},\Theta )}^{2}\left(\Theta \right) \end{array}$$
where $h(d,\vec{s},\Theta )$ is given by (13).

For our experiment, we generated N=1000 iid data points according to the Beta-MCA model with the settings mentioned above (see Figure 2A for the illustration of such data points). Next, we used the full EM procedure presented in Algorithm 1 to recover the ground-truth parameters (50 EM iterations were observed to be sufficient). To initialize matrices *W* and *V*, we randomly (uniform) chose H+1 data points and then computed the values of log(y) and log(1−y) and rearranged the results into two different matrices with D×(H+1) entries. The matrix obtained from the values of log(y) was used for the initialization of *W* and the other for the initialization of *V*. We also initialized $\pi_{h}$ values at 0.3. The results are then presented in Figure 2 where the background parameters are deliberately separated for the sake of a better visualization.

As it can be seen, the model extracted all bar patterns of the mean. All basis functions of the mean *M* have also learned traces of the checkerboard background. The reason for this is the maximum combination and the fact that the background is always present. As the non-bar pixels of the bar patterns are smaller than the background pixels (they have values of zero), the likelihood or free energy does not change for values between zero and the value of the background. Therefore, there is no reason for the algorithm to converge to any specific value, and learning stops once the value of the background pixel is reached. The variance basis functions ${\vec{\Sigma }}_{h}$ do seem to converge to patterns very different from the generating (i.e., ground-truth) patterns. At a closer inspection, also the variance patterns are optimal in the likelihood sense, however. Again, the maximum non-linearity explains the divergence from the original patterns: For any non-bar pixels of component means (with zero values), the background will always be larger. The variance of any non-bar pixel is consequently never used to generate a bar (is irrelevant for the likelihood value). The only variance values that are important, are consequently the variance values for those pixels that correspond to high mean values (together with the background values). For those pixels, the learned variance values estimate the ground-truth variances well. Nevertheless, the learned ${\vec{\Sigma }}_{h}$ values are observed to be slightly noisy that can be interpreted as the effect of finite sample size. Importantly, the model can distinguish between the two causes with the same mean value and different variances. In this context, Figure 2D illustrates the inference results for two unseen data points where each of them has only one of these two causes as the active cause. The model can then correctly infer the true active causes for these two data points (even though it still assigns a low probability for the activation of the other cause with the same mean value). Finally, we observed that the Beta-MCA algorithm estimates the generating $\pi_{h}$ values well.

Figure 2 shows the result of an execution of Algorithm 1 with a finally relatively high free energy value. When executing several runs with different initial conditions, we also frequently observed convergence to lower free energy values (convergence to local optima). When measuring the number of bars for the mean that can correctly be inferred after learning, we find an average of 8 bars out of 10 across different runs. However, as we can compute the free energy exactly, we are able to simply use the run with highest likelihood out of several runs. Alternatively, it would be possible to introduce additional annealing strategies (see, e.g., [28] for a concrete discussion).

Overall, the results approve the ability of the proposed Beta-MCA model and also the corresponding update Equations (Equation 19) and (Equation 20) in recovering (approximately) optimal model parameters for interval data.

### 5.2. Application to Medical Data

For the purpose of showing the application of the Beta-MCA model to medical data, a data set containing Common Audiological Functional Parameters (CAFPAs) is used. CAFPAs were introduced by Buhl et al. [9,23] as an abstract representation of the human auditory system which are independent of audiological tests performed for the respective patients. The data used here comprises CAFPA values determined by an expert survey, where leading clinical audiologists and physicians labelled the database from Hörzentrum Oldenburg, Germany, by indicating audiological findings, treatment recommendations, and CAFPAs for 287 single patients. The CAFPAs are defined on a continuous scale (the interval [0,1]) representing 0 as normal and 1 as pathological and describe different functional aspects of the auditory system: four CAFPAs are related to hearing thresholds in different frequency ranges, two CAFPAs are related to suprathreshold deficits below and above 1.5 kHz, and finally one CAFPA for each of the binaural hearing, neural processing, cognitive abilities and socio-economic components. We here excluded the socio-economic CAFPA as it describes the social environment of a patient rather than a physiological cause of hearing impairment. The data are then deliberately deidentified and here, as a first test, only CAFPAs for the audiological findings of high-frequency hearing loss and broadband hearing loss are used. This results in an amount of 124 data points with D=9 CAFPA values in which a number of 52 patients with high-frequency hearing loss, 26 with broadband hearing loss, 9 patients with both causes, and 37 with normal hearing are included. For the sake of computations, a small value of $1.0\times 10^{-10}$ is added (subtracted) to the CAFPAs with 0 (1) value.

We then used a model with H=2 (three causes including background) and applied Beta-MCA to learn the symptoms/causes relations between the two diseases and CAFPAs. We divided the considered data into two sets of training and test using 10-fold cross-validation and further assessed the performance of the proposed model in predicting the true active causes on the held-out data. That is, after learning parameters Θ of the model, we computed the following posterior probabilities for an unseen data point *y*:$$\begin{array}{c} S=\left\{\begin{array}{c} s^{\left(1\right)}=\left(0,0,1\right)\\ s^{\left(2\right)}=\left(1,0,1\right)\\ s^{\left(3\right)}=\left(0,1,1\right)\\ s^{\left(4\right)}=\left(1,1,1\right) \end{array}\right\}\;⟹\;\left\{\begin{array}{c} p(s^{\left(1\right)};y,\Theta )⟶\mathrm{probability}\;\mathrm{of}\;y\;\mathrm{being}\;\mathrm{healthy}\\ p\left(s_{1}=1;y,\Theta \right)⟶\mathrm{probability}\;\mathrm{of}\;s_{1}\;\mathrm{being}\;\mathrm{active}\;\mathrm{for}\;y\\ p\left(s_{2}=1;y,\Theta \right)⟶\mathrm{probability}\;\mathrm{of}\;s_{2}\;\mathrm{being}\;\mathrm{active}\;\mathrm{for}\;y\\ p\left(s^{\left(4\right)};y,\Theta \right)⟶\mathrm{probability}\;\mathrm{of}\;\mathrm{both}\;s_{1}\;\mathrm{and}\;s_{2}\;\mathrm{being}\;\mathrm{active} \end{array}\right. \end{array}$$
where S denotes the set of all possible state variables such that $s_{1}$ corresponds to the high-frequency hearing loss and $s_{2}$ to the broadband hearing loss.

We further considered three different data sets: Real CAFPAs, simulated CAFPAs and augmented CAFPAs. The first corresponds to training and testing on an amount of 124 real CAFPA values collected and labelled by the experts; the second denotes the synthetic data that we have generated for our purposes in this study (we will discuss its details further below); and the third corresponds to training on simulated CAFPAs and testing on real CAFPAs. Each of the cases will be discussed in the following.

In addition and for the sake of comparison, we also applied a probabilistic noisy-OR model (see Section A.2 for the details) on the binarized data set. To this, we compared each observable with an arbitrary and constant threshold α and set the binary output to 0 if the observed value is below the threshold and 1 otherwise. We repeated the experiments for several possible values of the binarization threshold, and the value to yield the best results found to be α=0.5. Receiver Operating Characteristic (ROC) curves are then depicted in Figure 3 (for all the three settings discussed above) where we used the scikit package *metrics* for computing the area under the curve (AUC) values.

Considering the middle column of Figure 3 (training and testing on real CAFPAs), it can be seen that the Beta-MCA can outperform the noisy-OR model on predicting the high-frequency hearing loss disease, but Beta-MCA performs worse than noisy-OR for broadband hearing loss. This can be seen as the effect of having very few amount of data points for training the model: Note that the Beta-MCA model has to estimate approximately twice as many parameters as the noisy-OR model. Consequently, the effect of finite sample size is more severe for the Beta-MCA. Therefore, in order to obtain a more informed comparison, a set of synthetic data, called simulated CAFPAs, is generated to be used for further analysis of the two diseases.

Simulated CAFPAs: we first learned the symptom statistics of the two diseases by fitting a Beta (and also a Gaussian for the sake of comparison) distribution to the data in which only one of the causes is active, i.e., data points with only the high-frequency hearing loss or the broadband hearing loss. Likewise, we fitted a Beta (and a Gaussian) distribution to the amount of 37 data points where none of the causes is active to learn the symptom statistics of a healthy patient. Doing so, we obtained the continuous disease profiles corresponding to the two diseases of high-frequency and broadband hearing losses and also of the normal case. The results of these disease profiles are further presented in Section A.3. In short, we observed that the Beta distribution is a better fit to the CAFPAs rather than Gaussian as the Beta distribution obtained higher log-likelihood values (see Figure A1 and Figure A2). This is in line with the results described in [9].

We further used the learned mean and variance parameters (obtained from fitting a Beta distribution) to generate N=1000 data points according to the Beta-MCA model (we refer to it as the simulated CAFPAs). The $\pi_{h}$ values were also computed by fitting a mixture of two Beta distributions to the data in which both diseases (high-frequency hearing loss and broadband hearing loss) are active.

We then considered the two models of Beta-MCA and noisy-OR and repeated the experiment above but this time using the (N=1000) simulated CAFPAs. The results are illustrated in Figure 3 (the first column from the left). Here, we observed that the Beta-MCA yields reliable results in inferring the symptoms/causes relations, and in predicting the true active causes for an unseen data point as it performs better in comparison to the noisy-OR model. Finally, for the third case, we trained the two models on simulated CAFPAs and tested the outcome on real CAFPAs (we refer to this case as the augmented CAFPAs). Considering Figure 3 (the first column from the right), we observed that additional data provided by simulated CAFPAs improves the performance of both models. Simulated CAFPAs are consequently a valuable form of data augmentation such that performance (in practice) can be improved. Additionally, we find that Beta-MCA is (as for simulated CAFPAs) now the preferable model: It achieves higher AUC values than noisy-OR with optimized threshold.

Especially for medical data, seemingly small differences of ROC curves can be very important in practice. As an example, we may again use COVID-19. For detections of COVID-19 infections it is very important to not miss an infection. Therefore, a classifier would have to operate at a high true positive rate (TPR, also known as sensitivity) ideally at >0.99. So, if the disease cause of the top-left plot of Figure 3 was COVID-19 instead of high-frequency hearing loss, then we could operate a Beta-MCA-based classifier above 99% TPR and it would produce less than 10% false positives (wrongly positive COVID-19 diagnoses). The noisy-OR-based classifier, on the other hand, would produce approximately 30% false positives in that case, i.e., more than three times as many patients would wrongly be diagnosed COVID-19 positive.

### 5.3. Feature Extraction—Natural Image Patches

To explore other applications and to investigate scalability, we trained the proposed Beta-MCA model on a set of N=100,000 natural image patches chosen randomly from the van Hateren database [64] (compare, e.g., [26,28,31,50]). Here we used D=8×8 image patches with pixel intensities linearly rescaled to fill the interval [0.1,0.9]. After rescaling, we added a small amount of Beta noise to the image patches (which increased stability). On these data points, we trained Beta-MCA models with H=100 latent components for 500 EM iterations. We were particularly interested in how learned data representations varied when transitioning from a model with a global scalar variance parameter $\sigma^{2}$ to a model with individual variances per latent component $\Sigma_{dh}^{2}$ (note that we here dropped the dependency on parameters Θ for the sake of readability). Consequently, we trained two different models: one with a dictionary *M* for component means and a global variance $\sigma^{2}$ (as in preliminary model (Equation 1)–(3); see Section A.1 for details about the corresponding M-step update rules) and one with individual dictionaries for component means and component variances (as described in Section 3).

As the size of the Beta-MCA model applied here did not allow for exact evaluations of posterior probabilities, we used the EEM approach to approximate E-steps (see Section 4.2; we used 60 variational states per subset $K^{\left(n\right)}$ for the EEM algorithm). To initialize parameters Θ of the two models, we applied the following procedure: component means ${\vec{M}}_{h}^{\mathrm{init}}$ were initialized with the mean of the data points plus a small amount of Gaussian noise, the prior components $\pi_{h}^{\mathrm{init}}$ and also the values of $\sigma^{\mathrm{init}}$ and ${\vec{\Sigma }}_{h}^{\mathrm{init}}$ were uniformly randomly sampled from the interval (0,1). We kept the variance parameters $\sigma^{2}$ and $\Sigma^{2}$ constant at their initial values during the first 30% of the iterations and only updated parameters *W* and $\vec{\pi }$ (we found this to lead to a more stable convergence behaviour of the algorithm).

The results are then presented in Figure 4. Looking at the generative fields (GFs) corresponding to the component means ${\vec{M}}_{h}$ learned with the Beta-MCA model with scalar variance parameter, we observed many globular fields, some elongated fields and some gratings (Figure 4A). For the variance, we inferred the single globular value of σ=0.086. For the model with individual component variances, the diversity of GFs increased, with many elongated fields (with different location and orientation), gratings, and with large globular fields (Figure 4B, top). A crucial qualitative difference of the model with individual variances is, of course, the additional GFs for the variances (Figure 4B, bottom). The variances allow for increased flexibility in terms of intensity variations of elongated fields (for many elongated fields variances are high where the corresponding mean is high). GFs for variances also allow for more intricate modelling of image structure, however. As can be observed, for some fields (e.g., bottom right) variance encoding allows for shape variations rather than intensity variations: *p* = only at the transition of high to low mean values are variances high, which means that variations of the precise shape modelled by a component are allowed and modelled.

In terms of free energy, the Beta-MCA model with individual component variances achieved a better fit to the data compared to the model with global variance parameter (Figure 4E): the corresponding final free energy value was **82.03** for the model with individual variances compared to **70.31** achieved by the model with global variance parameter. While a higher free energy can be expected because of more parameters of the model with individual variances, the higher free energy can be taken as a confirmation that the encoding of more intricate image structure in Figure 4B corresponds to a more refined model of image patch structure.

To obtain more intuition about the learning results, we also applied the trained models to generate new data points which can then visually be compared to the image patches used for training. Consistent with the higher free energy values, we observed that the data points generated using the model with individual component variances appear to be more similar to the image patches used for training (Figure 4D,F,G).

Finally, we repeated the experiment several times using different choices of hyperparameters. For instance, we ran the two algorithms again with H=200 and in another run with H=100 and D=16×16. In all different cases, we observed similar results both qualitatively and quantitatively. In particular, the model with individual component variances achieved consistently the highest final free energy value.

Consistent with the higher free energy value, the current experiment illustrates that the variance components learned by the proposed Beta-MCA (with double dictionaries) provide (in contrast to the Beta-MCA with scalar variance and other conventional LVMs) more accurate information on how first and second order statistics of data are combined. That generative models with non-linear combinations of latents are particularly well suited to model, for instance, occlusion effects in natural image patches (as has been argued previously, e.g., [26,27,28,31]). Besides, generative models with individual component variances have been applied to natural image patches before very recently [33,50]. All of these previous works used generative models with either Gaussian or Poisson noise assumptions, however. Considering Figure 4, this paper is, to the knowledge of the authors, the first to report results on images of a non-linear generative model with a Beta noise assumption and individual component variances.

### 5.4. Denoising

Finally, we study the concrete example of denoising an image corrupted by Beta noise. Many algorithms have been developed so far for the removal of additive white Gaussian noise [65,66,67,68,69,70,71,72] which, to some extent, can also be applied for denoising of an image corrupted by Beta noise. However, we here show that the proposed Beta-MCA model is better suited for this task as it assumes a Beta observation noise. As an example, we used the ’house’ image [65] which we rescaled from its original interval of [0,255] to lie in the interval [0.2,0.8]. The image was then corrupted by Beta noise with a standard deviation of 0.3 (we considered a homoscedastic noise as it is a common procedure for the denoising experiments), and further segmented into patches of size D=12×12. Therefore, we again used the Beta-MCA model with a global variance parameter. We then applied the Beta-MCA model on the noisy data without any further pre- or post-processing and performed 1000 training iterations with H=512 components. We also applied variational approximation as for the previous experiment (with the same settings) in order to scale the EM algorithm.

Note that the performance of any denoising algorithm will depend on many aspects of the model. This includes the used approximate inference approach for the E-step, the averaging and inpainting algorithms, or methods to avoid local optima during learning. We here did not use annealing to avoid local optima, and also did not fine-tune approximate inference or the averaging algorithm to improve the denoising performance. In general, such fine-tuning or even automatic optimization of hyperparameters can significantly further improve performance.

After parameter optimization, we followed e.g., [45] and used the encoding learned by the model to estimate the underlying image for which we employed the estimator:(26)$${\left({\vec{y}}_{\mathrm{estimate}}^{\;(n)}\right)}_{d}={\langle {\bar{\mu }}_{d}\left(\vec{s},\Theta \right)\rangle }_{q^{\left(n\right)}}.$$

In addition to the Beta-MCA model mentioned above, we further applied MCA [26,31] and spike-and-slab sparse coding [44,73] (SSSC) models each with H=512 components and 1000 EM iterations, and also the sparse 3D transform-domain collaborative filtering [65] (BM3D) to the corrupted image. All these three approaches are tailored to Gaussian observation noise but can directly be applied here as we use continuous data. Amongst these approaches, the first two are based on generative models and exploit an EM approach for training. The BM3D algorithm is not a generative model but a strong-performing, highly engineered denoising approach that produces competitive results for the removal of the additive white Gaussian noise.

Figure 5 depicts the results of the denoising experiment where the reconstruction performances are compared using the standard measures of peak-signal-to-noise-ratio (PSNR) and mean squared error (MSE). As can be seen, Beta-MCA obtained the best reconstruction performance in terms of both PSNR and MSE values. Still the two measurements do not best reflect the differences between the restored images in Figure 5 as visual inspection may arguably favor the results of the Beta-MCA (in comparison to the other three approaches) more than that the raw PSNR and MSE values suggest. The results may be taken to confirm both scalability and effectiveness of the Beta-MCA model for continuous interval data.

Finally, in order to assess the performance of Beta-MCA if the noise is not Beta distributed, we repeated the experiment but this time using Gaussian noise. To this, we again considered the house image (Figure 5a) and corrupted the image by additive Gaussian noise with σ=10. We then rescaled the corrupted image to the interval [0.01, 0.99] and applied the Beta-MCA model, (Gaussian-)MCA and SSSC to denoise the image (rescaling enables the application of all models). Concretely, we performed 1000 EM iterations for each of the three models and used the same settings as for the previous experiment. The results are presented in Figure 6. As the considered data is Gaussian distributed, we observed that the two models of MCA and SSSC, which are tailored to Gaussian noise, perform best. The performance of Beta-MCA is almost as high as the performance of MCA, though, which is evidence for the flexibility of the Beta distribution. The SSSC approach is clearly the most competitive, which may not be surprising as the model (A) assumes the correct type of noise in this case, and (B) has a relatively flexible prior which is advantageous for denoising. On the other hand, when the noise is Beta-distributed and the SSSC noise assumption is thus not correct, Beta-MCA is observed to produce significantly better denoising results (Figure 5).

## 6. Discussion

We here investigated a latent variable model (LVM) for observables with values in a finite interval. Interval data is commonly encountered, e.g., in medical data recordings [9,10,11,74] and other domains (we, e.g., considered intervals of [0,255] for gray level images). A natural parametric distribution for interval data is the Beta distribution. However, inference and learning of latent variable models with Beta-distributed observables is very challenging. To infer latent causes, any interval data could, in principle, be binarized in order to apply standard models such as noisy-OR-like nets. Alternatively, we here endeavored in modelling interval data directly. We took first steps in this novel direction by basing the investigated LVM approach on the maximum non-linearity [28]. Related modelling approaches to address non-binary or non-Gaussian types of observation statistics in other contexts is, e.g., work by Vergari et al. [42]. Their approach focuses on discovering the statistical data type of the model automatically, and to provide a number of automatic data analysis tasks (also see [41]). While being applicable to interval data (and more generally), the approach does not provide a model for the combination of means and variances which we believe is of importance, e.g., for medical data. Another related approach by [37] uses a sparse coding model with observables following, in principle, any distribution of the exponential family. The interaction between causes and observables of this approach is chosen to make inference easy, however, rather than modelling actual causes-observable relations in the data. Furthermore, neither experiments for Beta distributions were discussed, nor were variance combinations considered. Other similar studies are [38,39,75] which are all, in contrast to the proposed model here, based on a linear link function. Except of the standard Gaussian, distributions depending on two parameters are avoided in the above discussed approaches, as substantial changes to standard LVMs are required in this case. Often not only the actually realized algorithms are constrained to one-parameter distributions but also the derivations assume one-parameter distributions and sometimes only the distributions where one element of their sufficient statistics is proportional to *y* (e.g., [38]). Neither is the case for the Beta distribution: the Beta distribution is a two-parameter distribution of the exponential family and none of its sufficient statistics is proportional to the mean. While the Beta distribution is more complex than observable distributions previously considered for LVMs, we here argued that appropriate modelling of both mean as well as variances within an LVM approach can result in important advantages compared, e.g., to approaches using a binarization followed by an LVM (as noisy-OR) with a one-parameter distribution. Using recent theoretical results [33] for LVMs with a maximum combination of binary causes, we derived an LVM model that could model the relation between latents and observables w.r.t. the mean values of the Beta distribution as well as w.r.t. the distributions’ variances.

The novel Beta-MCA model we considered thus allows for inferring multiple causes (a property that it shares, e.g., with noisy-OR Bayes nets and sigmoid belief nets) based on two sets of generative fields, one set containing the means associated with each cause and one set (with equally many parameters) that models the variances associated with each cause. Such individual variances per latent are known in the context of Gaussian mixtures to carry important information [76]. For medical data, we have here argued that the same is true for a multiple-cause model with individual variances. Disease profiles can be obtained that are also instructive about the variance of a symptom, and numerical experiments on augmented data collected from medical experts show that classifiers based on interval data can perform more strongly than standard classifiers based on binary data. Applications to image data, furthermore, show that the method can be scaled (we used up to *N* = 100,000 with up to H=100 on data points with dimension D=8×8). Features extracted by the novel algorithms are qualitatively different with variances modelling feature variations; and quantitative evaluations of denoising for data with Beta noise shows the capabilities of the LVM model for a typical task addressed by LVMs.

Of course, the development of an LVM which models means and variances of the Beta distribution is much more challenging than a binarization of the data followed by standard noisy-OR, or compared to assuming Gaussian distributions. We believe, however, that tackling such challenges is important to improve inference based on LVMs for interval data—and it is worth the effort as improvements can be very important, e.g., for medical data. We have here shown that an effective and scalable LVM based on Beta distributed observables can indeed be developed—which makes the presented model unique in these respects.

## Figures and Tables

**Figure 1 entropy-23-00552-f001:**
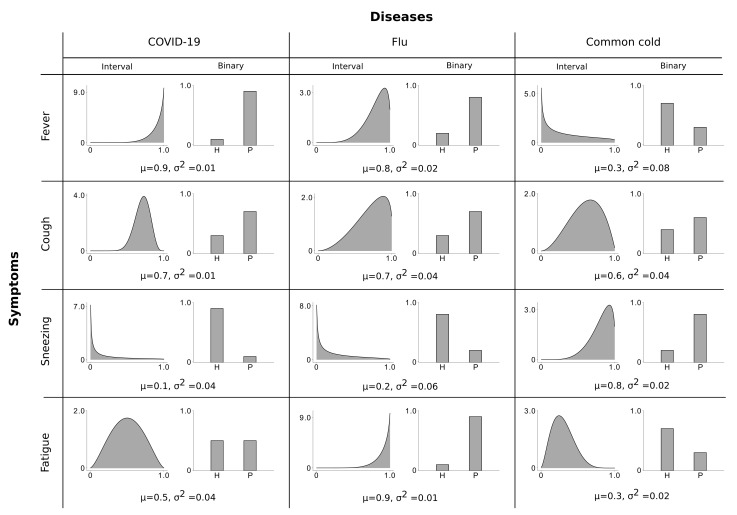
A disease profile illustrating the differences between a binary and a continuous interval encoding of the observables. Here, the interval [0,1] is used to model the severity of the symptoms where for 0 the symptom is completely absent while 1 models the most severe (most pathological) state of the symptom. For binary symptoms, the states ’H’ and ’P’ refer to the ‘healthy’ and ‘pathological’, respectively. From the figure, it can be seen that a continuous interval encoding models a higher order statistics of the data and could provide more information compared to a binary encoding. For instance, the statistical properties of the symptom “cough” for “COVID-19” and “flu” can be distinguished if an interval encoding is used, while a binary encoding makes this symptom identical for the two causes.

**Figure 2 entropy-23-00552-f002:**
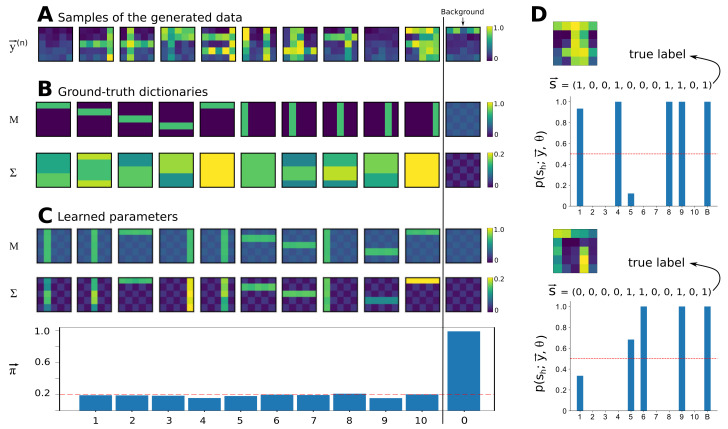
(**A**) A total of 11 examples of input data. (**B**) Ground-truth mean and variance dictionaries used to generate data. As illustrated, generative fields $s_{1}$ and $s_{5}$ have the same mean value but different variances. (**C**) Learned mean and variance dictionaries together with learned $\pi_{h}$ values after 50 EM iterations. As observed, the model is able to learn all parameters with a fairly good precision. Importantly, the learned background patterns and also learned variance dictionaries reveal the robustness of the model in training interval data. (**D**) Two example data points that are chosen such that cause $s_{1}$ is active for one of them (the upper data point) and $s_{5}$ is active for the other (the lower data point). The posterior values $p(\vec{s};\vec{y},\Theta )$ given the learned parameters are then computed for these two data points and the results are shown accordingly. As it can be seen, the model successfully distinguishes between the two causes with the same mean value and different variances and infers the active causes correctly. Observe that the model still assigns a low probability to cause $s_{5}$ for the upper case and to cause $s_{1}$ for the other.

**Figure 3 entropy-23-00552-f003:**
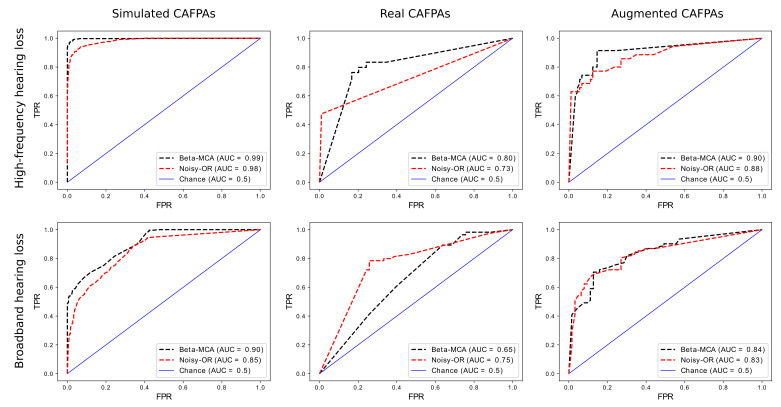
Illustration of the ROC curves presenting the results of the Beta-MCA and noisy-OR models trained on simulated, real and augmented CAFPAs (see text for details). The two models of Beta-MCA and noisy-OR are distinguished with black and red colors respectively. Moreover, the blue line denotes the prediction results corresponding to tossing a coin (each disease have 50% chance to be active). The appeared AUC values reveal the beneficiary of the Beta-MCA in 5 out of 6 cases.

**Figure 4 entropy-23-00552-f004:**
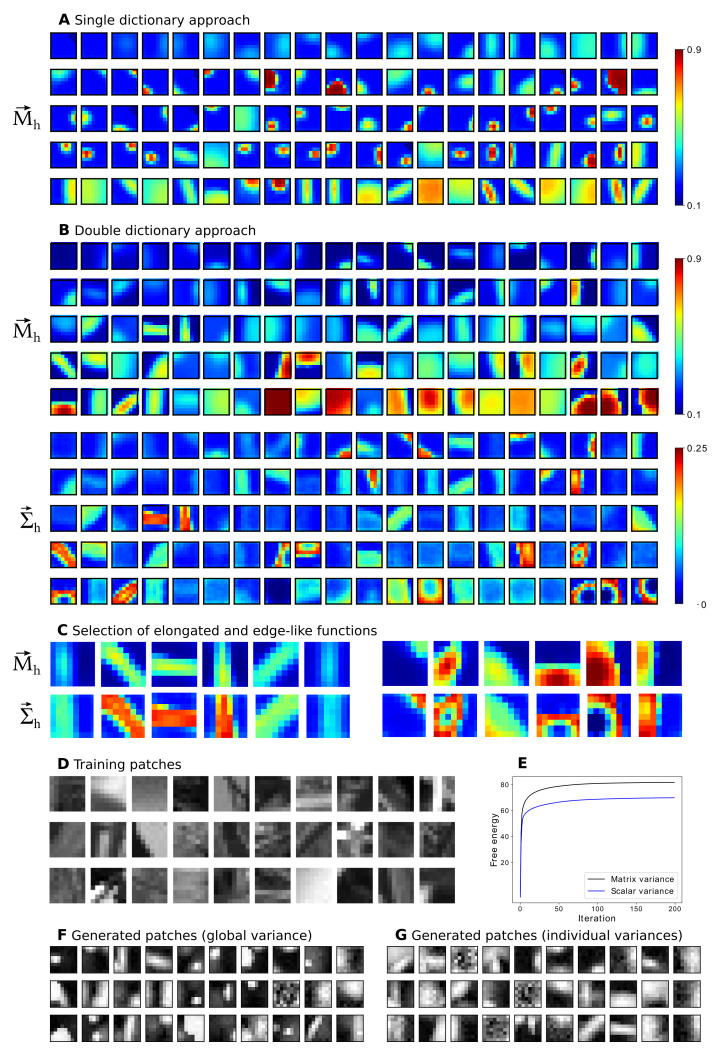
Feature extraction. (**A**,**B**) Parameters ${\vec{M}}_{h}$ and ${\vec{\Sigma }}_{h}$ of the Beta-MCA generative model trained with (**A**) a single dictionary (a scalar variance $\sigma^{2}$) and (**B**) individual dictionaries for component means and component variances. Displayed are H=100 generative fields sorted from left to right and from top to down based on their prior values, $\pi_{h}$. The top left corresponds to the background which has a prior parameter close to 1. Each of the component variances in the lower part of panel (**B**) corresponds to the respective component means in the upper part of panel (**B**). (**C**) Examples of (left) elongated and (right) edge-like functions learned with the double dictionary approach. (**D**) 30 training patches. (**E**) Evolution of the free energy for Beta-MCA model for both single and double dictionary approaches. Here, only the first 200 (out of 500) iterations are depicted. (**F**,**G**) 30 examples of the patches generated according to the Beta-MCA trained with (**F**) scalar variance and (**G**) matrix variance. We here deliberately reduced the amount of noise in the generation process for the sake of visualization. Additionally, in all the above cases, we linearly scaled the learned values to fill the corresponding color range in the gray (color) space.

**Figure 5 entropy-23-00552-f005:**
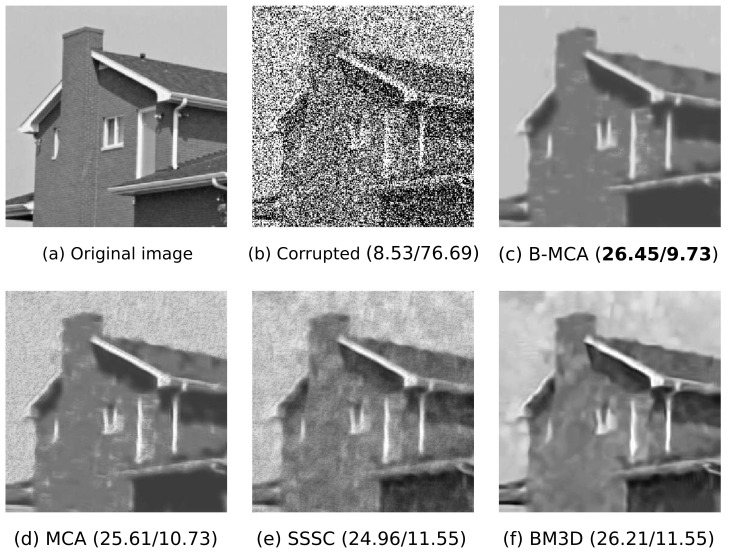
Denoising of the house image corrupted by Beta noise with $\sigma^{2}=0.3$. (**a**) is the original noiseless image, and (**b**) the image with Beta-noise. (**c**–**f**) depict reconstructed images obtained by applying four different data models. Numbers in parentheses are the corresponding PSNR (in dB) and MSE values (first number is PSNR, second number is MSE). As it can be seen, the Beta-MCA (B-MCA) model improves on the other three approaches, which is consistent with the here used Beta noise. See text for further details.

**Figure 6 entropy-23-00552-f006:**
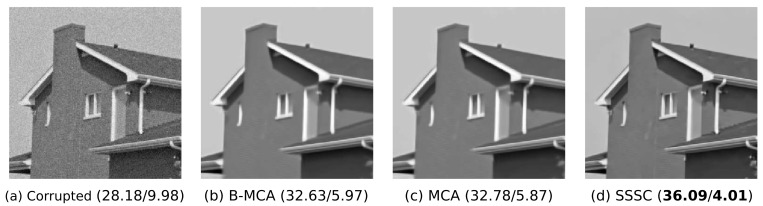
Denoising of the house image corrupted by Gaussian noise. (**a**) Image with added Gaussian-noise. (**b**–**d**) depict reconstructed images obtained by applying three different data models. Numbers in parentheses are the corresponding PSNR (in dB) and MSE values (first number is PSNR, second number is MSE). As illustrated, denoising with Beta-MCA (B-MCA) and (Gaussian-)MCA works similarly well but the SSSC model shows the best performance. See text for further details.

## Data Availability

Not Applicable.

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
