# Peer review of "Inference and Learning in a Latent Variable Model for Beta Distributed Interval Data"

_entropy, 2021, doi:10.3390/e23050552_

Round 1

Reviewer 1 Report

Overall:

This paper is about latent variable models for interval data. The authors carefully build up a mathematical model for interval-data, and tackle the issues related to this model using variational EM (both trying with exact  and approximate E-step). The development of the model is indeed challenging, and I am not aware of anyone having tackled this in a similar manner. The results presented appear sound. 

The modelling is backed up by a sufficient set of experiments, both doing more-or-less standard tests as well as examining their own medical data set.

The presentation is mostly clear, and written in a very readable English.

Overall I have no problems suggesting the paper is accepted after minor modifications as outlined below. 

Detailed comments:

Page 4, line 143: "... (which is ASSUMED TO BE the same for all..."

Page 6, line 213: Maybe add that the "standard" parameterization for the Beta distribution (using \alpha and \beta) is in fact the same as the natural parameters in the exponential family representation. 

Page 7, line 215: I first didn't see / take in the *arg*max in eq 12 (as I was in the mindset of Eq 3), so that h is later used as an indexing into the V and W tables. It is stated clearly in "line 228.5", but I think it would be better if you push this a bit forward.

Page 7, Line 219: It is not completely clear at this point why we need to go between the different representations, and therefor have to have to assume \eta_1 \neq \eta_2. Maybe say something about that here?

Page 8, Eq 14 + 15: So, these equations only work when \eta_1 \neq \eta_2. I am not too concerned about the assumption if it is only the single point for which \Psi doesn't exist, but can you elaborate on the behaviour as |\eta_1 - \eta_2| \goesto 0? Is it "ill-behaved" (something like 1/x as x goes to 0), or is it "nice" (like e.g. sin(x)/x as x goes to 0)? Basically I hope you can say something about the consequences of this assumption.

The experiments are nice, appear fair, and are enlightening. Two comments, though:

  • Why isn't the model (1)--(3) tested on the medical data in 5.2?
  • In Sec 5.4 you establish that under the generative model of Beta-MCA, Beta-MCA is indeed better than candidates assuming a different generative model. This is fine, and maybe also expected. However, in practice I guess you will rarely know the generative process, and I was wondering about the other way around. I therefore would welcome an additional experiment with Gaussian noise. What happens now?

Reviewer 2 Report

In this work the authors aim at generalizations of standard noisy-OR models towards models with values that lie within an interval. The study is clearly presented and founded. The experiments are described in detail (though it would have been desirable a larger experimental study with more datasets). The bibliography is quite complete and the final appendixes are appropriate.

Taking all of this into account, I think that this work can be published in this version in Entropy. 

Author Response

We thank the reviewer for appreciating our work. We have just uploaded a revised version which includes a novel control experiment (Figure 6) and some minor changes asked for by reviewer 1.